# DASGAN - Joint Domain Adaptation and Segmentation for the Analysis of Epithelial Regions in Histopathology PD-L1 Images

A. Kapil[1], T. Wiestler[1], S. Lanzmich[1], A. Silva[1], K. Steele[2],
M. Rebelatto[2], G. Schmidt[1], N. Brieu[1]

[1] Definiens, Munich, Germany
[2] AstraZeneca, Gaithersburg, USA

**Abstract.** The analysis of the tumor environment on digital histopathology slides is becoming key for the understanding of the immune response against cancer, supporting the development of novel immuno-therapies. We introduce here a novel deep learning solution to the related problem of tumor epithelium segmentation. While most existing deep learning segmentation approaches are trained on time-consuming and costly manual annotation on single stain domain (PD-L1), we leverage here semi-automatically labeled images from a second stain domain (Cytokeratin-CK). We introduce an end-to-end trainable network that jointly segment tumor epithelium on PD-L1 while leveraging unpaired image-to-image translation between CK and PD-L1, therefore completely bypassing the need for serial sections or re-staining of slides. Extending the method to differentiate between PD-L1 positive and negative tumor epithelium regions enables the automated estimation of the clinically relevant PD-L1 Tumor Cell score. Quantitative experimental results demonstrate the accuracy of our approach against state-of-the-art segmentation methods.

**Keywords:** Domain Adaptation · Semantic Segmentation · Digital Pathology · Generative Adversarial Networks

## 1 Introduction

Histopathology is key to many clinical decisions taken in oncology, based on the visual quantification of biomarkers on stained slides of suspected tumor tissue. In a clinical setting, the PD-L1 Tumor Cell (TC) score for Non Small Cell Lung Cancer (NSCLC) is for instance predictive of response for patients treated with an anti-PD1/PD-L1 checkpoint inhibitor therapy [11]. Several exploratory studies have moreover shown that both tumor immune contexture [4] and epithelial immune cell infiltration are predictive of patient prognosis [1]. All these examples rely on an accurate segmentation of the epithelial compartment. The non-specificity of the PD-L1 staining, which includes epithelial regions but also immune cells and necrotic regions (cf. Fig. 1b) makes this task challenging. This difficulty, together with the demonstrated performance of deep learning methods in digital pathology image analysis [7, 3, 6] leads us towards this set of methods,

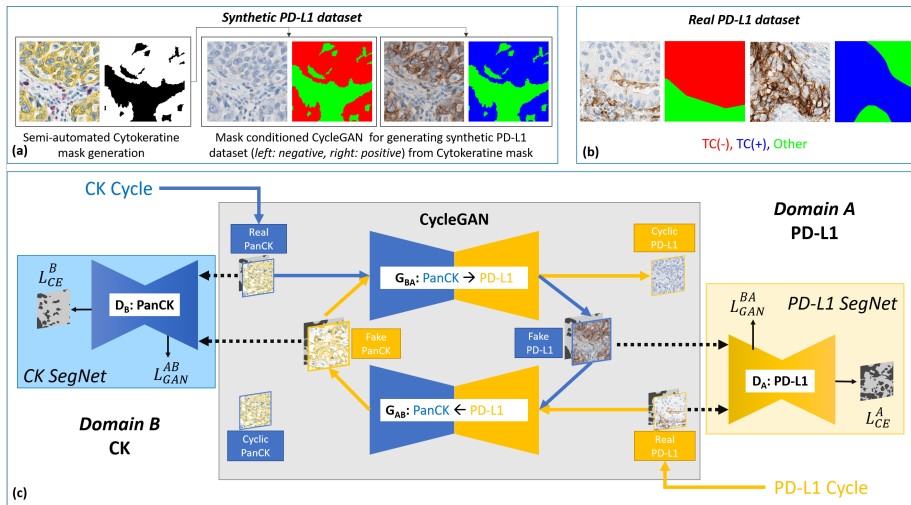

**Fig. 1.** Synthetic (a) and real (b) PD-L1 datasets generated from the semi-automated segmentation of CK images and manual annotations, respectively. (c) DASGAN model for joint domain adaptation and semantic segmentation. NB: the two cycle losses between the real and the cyclic images are not displayed for clarity purposes.

and more particularly towards deep semantic segmentation networks [9]. The prerequisite dataset of boundary-precise manual annotations is, however, time-consuming and costly to generate. Here, and because CK labels epithelium, we semi-automatically build the prerequisite dataset on CK images using coarse manual input combined with simple heuristic segmentation rules. By transferring the segmentation masks and the CK images into the PD-L1 stain domain using unpaired image-to-image translation, we generate a synthetic CK-based PD-L1 dataset which is merged with manual annotations on true PD-L1 images and used for training the PD-L1 epithelial semantic segmentation network.

We exploit recent advances in generative adversarial networks (GANs), especially of unpaired image-to-image translation using CycleGAN [15] as used for the normalization of HE images [12]. This makes the transformation of CK images into synthetic PD-L1 images possible without the need for serial sections nor re-staining [13]. Here, we present what is to our knowledge the first application of domain adaptation based semantic segmentation in the field of digital pathology. Other studies described similar ideas for medical image analysis, e.g. to convert CT images into synthetic MRI images and train a segmentation network on both real and synthetic MRI images[2, 5], but follow a two-step methodology. Instead, we introduce an end-to-end trainable network (cf. Fig. 1c) named DASGAN (Domain Adaptation and Segmentation GAN) that jointly performs unpaired image-to-image translation and semantic segmentation. We show the superiority of the introduced network against (i) networks trained only on manual annotations of real PD-L1 images and (ii) networks trained separately for domain adaptation and for semantic segmentation.

## 2 Methods

The good specificity of CK staining makes the segmentation of epithelial regions in CK images possible using color deconvolution followed by Otsu thresholding and closing morphological operations (cf. Fig. 1a). While the newly introduced network for joint unpaired domain adaptation and semantic segmentation theoretically makes it possible to combine manual or automated annotations from any two stain domains and independent cohorts, we apply it here to transfer images from the CK domain to PD-L1 domain and to leverage the epithelial segmentation masks in the CK domain as annotations in the PD-L1 domain.

### 2.1 CycleGAN

Two generators $G_{BA} : \mathcal{X}_B \to \mathcal{X}'_A$ and $G_{AB} : \mathcal{X}_A \to \mathcal{X}'_B$ are trained to synthesize samples in domain $A$ (PD-L1) from real samples in domain $B$ (CK) and vice versa. Two discriminators $D_A$ and $D_B$ are trained in opposition to identify synthetic from real samples in the two domains. The parameters of the two discriminator and two generator networks are learned in an adversarial manner following a min-max game on the two adversarial losses $\mathcal{L}_{GAN}^{AB}$ and $\mathcal{L}_{GAN}^{BA}$:

$$\min_{G_{AB}} \max_{D_B} \mathcal{L}_{GAN}^{AB} := \mathbb{E}_{x_B \sim \mathcal{X}_B} \log(D_B(x_B)) + \mathbb{E}_{x_A \sim \mathcal{X}_A} \log(1 - D_B(G_{AB}(x_A))) \quad (1)$$

$$\min_{G_{BA}} \max_{D_A} \mathcal{L}_{GAN}^{BA} := \mathbb{E}_{x_A \sim \mathcal{X}_A} \log(D_A(x_A)) + \mathbb{E}_{x_B \sim \mathcal{X}_B} \log(1 - D_A(G_{BA}(x_B))) \quad (2)$$

The necessity of having image pairs for image translation between $A$ and $B$ is bypassed using a cycle consistent loss $\mathcal{L}_{cycle}$ [15]. The cycle loss is defined to prevent mode collapse of the two GAN models and to constrain the invertability of the translated domains, based on the translation of the synthesized samples $x'_B = G_{AB}(x_A)$ and $x'_A = G_{BA}(x_B)$ back to their original domains $A$ and $B$:

$$\mathcal{L}_{cycle} := \mathbb{E}_{x_A \sim \mathcal{X}_A} \|x_A - G_{BA}(x'_B)\| + \mathbb{E}_{x_B \sim \mathcal{X}_B} \|x_B - G_{AB}(x'_A)\| \quad (3)$$

### 2.2 DASGAN

Following the auxiliary classifier GAN (AC-GAN) model [10], we extend the CycleGAN model [15] to obtain segmentation maps as auxiliary from the two discriminators $D_A$ and $D_B$ (cf. Fig. 1c). We condition the input images of $G_{AB}$ and $G_{BA}$ with the respective ground truth segmentation class mask by concatenating the mask across the input image channel axis. The respective concatenated volumes go through a series of transformations by $G_{AB}$ and $G_{BA}$ to produce synthetic images in the respective target domains $B$ and $A$. The two discriminator networks are extended to predict pixel-wise class probability maps in addition to predicting the correct source of image. To this end, and to propagate the class specific information through to the generator, a segmentation loss is introduced to the discriminator in addition to the original adversarial loss:

$$\mathcal{L}_{seg} := \mathcal{L}_{CE}(y_A^{true}, y_A^{pred}) + \mathcal{L}_{CE}(y_B^{true}, y_B^{pred}) \quad (4)$$

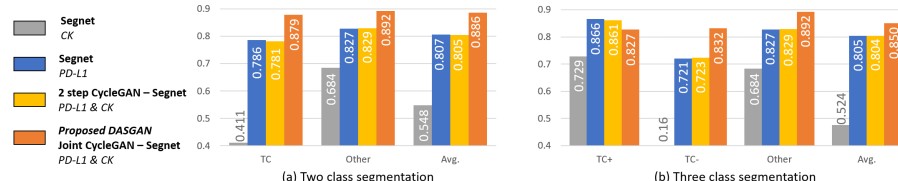

**Fig. 2.** Segmentation accuracy on the unseen test set, for the three baseline models (in gray, blue and yellow) and for the proposed DASGAN model (in orange) under the condition (i) of low availability of manual annotations on real PD-L1 images. F1 scores are reported for each class of interest - epithelium (TC), epithelium positive (TC+), epithelium negative (TC-) and Other, together with their average score Avg. in both scenarios of epithelium detection and replication of TC score.

where $\mathcal{L}_{CE}(y^{true}, y^{pred}) = -\sum y^{true} \log(y^{pred})$ denotes the categorical cross-entropy loss and where $y^{true}$ and $y^{pred}$ correspond to the ground truth and the predicted label maps respectively. This results in the following loss for the proposed joint domain adaptation and semantic segmentation DASGAN model:

$$\mathcal{L} := \mathcal{L}_{GAN}^{AB} + \mathcal{L}_{GAN}^{BA} + \lambda_1 \mathcal{L}_{cycle} + \lambda_2 \mathcal{L}_{seg}, \tag{5}$$

with $\lambda_1 = 10$ and $\lambda_2 = 1$ weighting the losses associated with the cycle constrain and the segmentation auxiliary task respectively. The proposed DASGAN model is used, at training training time, to leverage annotations on CK stained images for the segmentation of epithelial regions in PD-L1 stained images. While only the discriminator $D_A$ is employed at prediction time, the use of a symmetric discriminator $D_B$ ensures the balancing of the two counterplaying GAN networks.

### 2.3   Extension to Tumor Cell scoring

To differentiate between PD-L1 positive and PD-L1 negative tumor epithelial regions as requisite for the calculation of the TC score, we perform three-class pixel-wise mask conditioning. We transform each CK binary segmentation mask into two examples of both a PD-L1 negative and a PD-L1 positive epithelial masks that results in two versions of same CK image (c.f. Fig. 1(a)). Given a CK binary segmentation mask, a PD-L1 negative epithelium mask is built by giving the labels 0 and 1 to the non-epithelium and the epithelium regions respectively. This conditions the generator to yield a PD-L1 negative image. Similarly, a PD-L1 positive epithelium mask is built from the same CK mask by giving the respective labels 0 and 2 instead. This conditions the generator to yield a PD-L1 positive image from the same CK image.

After describing the training, validation and test datasets as well as the network architectures, we present results of quantitative evaluation against pathologist manual annotations for the two problems of (i) epithelial segmentation and (ii) PD-L1 positive and negative epithelial region detection.

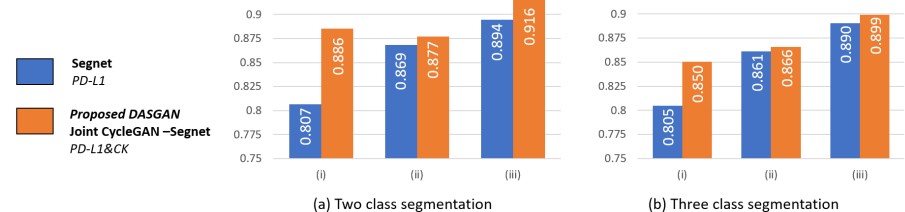

**Fig. 3.** Segmentation accuracy (avg. f1-score) on the unseen test set, for the baseline model trained only on real PD-L1 samples (blue) and the proposed DASGAN model trained on both real and synthetic PD-L1 samples (orange), for increasing availability (i)-(ii)-(iii) of manual annotations on real PD-L1 images.

## 3 Experiments and Results

### 3.1 Cytokeratin and PD-L1 Datasets

The training set consists of $N_{CK} = 56$ CK stained whole slide images (WSI) of NSCLC samples and of $N_{PD-L1} = 69$ WSIs of the same indication and stained with the SP263 PD-L1 clone. The CK images and the PD-L1 images are unpaired and come from two independent patient cohorts. To ensure purity of the samples generated on the CK stained slides, the samples are created on tumor regions delineated by pathologists and regions with non-specific staining are discarded. Because this manual input is provided at a macroscopic scale (1x or 2x), the associated effort is minimal compared to the annotation of fine epithelium structure at high resolution (20x). Positive ($TC+$) and negative ($TC-$) epithelium regions were partially delineated at high resolution on PD-L1 images, together with non-epithelium regions (e.g. immune, necrotic, and stromal regions). Patches of $128 \times 128$ pixels are uniformly sampled from the annotated regions on the PD-L1 stained slides and from the detected epithelium regions on the CK stained slides, using a $10\times$ resolution ($1\mu m$/px). The validation set, similarly generated from 28 partially annotated PD-L1 stained WSIs, is used for selecting the model maximizing the segmentation f1 score. The test set, consisting of 106 fields of view ($500 \times 500\mu m$) selected from 25 PD-L1 stained WSIs is densely annotated by pathologists. It is solely employed for quantitative evaluation of the segmentation accuracy and was selected to cover a high variability of different cancer types (adeno, squamous) and growth pattens (acinar, papillary and solid). To study the impact of $N_{PD-L1}$ on the segmentation accuracy, we report results with three different configurations for the training and validation sets: (i) 44K patches from $N_{PD-L1} = 22$ slides, (ii) 103K patches from $N_{PD-L1} = 49$ slides and (iii) 149K patches from $N_{PD-L1} = 69$ slides, all patches from (i) being included in (ii) and those of (ii) in (iii). The CK-based training and validation sets, as well as the PD-L1 test set remain unchanged in these experiments. Note that, a thorough quantification the quality of the domain adaptation is not in the scope of this work, which is more towards its final impact on semantic segmentation results.

## 3.2   Network architectures

The architectures of the two generators are similar to that in the original Cycle-GAN paper [15] with minor modifications: we concatenate the input images and the segmentation mask. For the two discriminators, weights between the prediction of the source distribution and of the semantic segmentation posterior maps are shared in the first three convolutional layers and the branch for semantic segmentation extended to include three resnet blocks and three deconvolutional layers. Spectral normalization [8] and self-attention blocks [14] are added in the discriminators and generators to increase training stability and to model long structural dependencies respectively. Network definition, training and inference are performed using the Tensorflow library. All models are trained on a single Nvidia V100 GPU with 32GB of memory and Adam optimization performed for both the generators (lr=1e-4, beta1=0.5) and the discriminators (lr=5e-4, beta1=0.5) for 150k iterations. Because the same architecture $D_A$ is used by all networks for segmentation, the prediction time is the same for all networks: 0.08 sec for $512 \times 512$ pixels is measured on Nvidia K80 GPU.

## 3.3   Segmentation performance

Segmentation accuracy is reported on the unseen test set. We first consider the configuration (i) corresponding to a relative shortage of manual annotations. As illustrated in Fig. 2, the proposed DASGAN outperforms the two models trained solely on real or synthetic PD-L1 images as well as the two-step model trained on real and synthetic PD-L1 images. Mean f1 scores of $f_1 = 0.886/0.850$ are reported for the DASGAN on the binary problem of epithelium detection and on the three class problem of positive and negative epithelial regions respectively. Surprisingly, the two-step approach does not improve the segmentation results ($f_1 = 0.805/0.804$) compared to training only on real PD-L1 images ($f_1 = 0.807/0.805$). A possible explanation is that, while the transformation between the two stain domains is fixed in the two-step methodology, the DASGAN enables the domain transfer network to be optimized with the objective to not only generate realistic PD-L1 images but also to ensure that the generated images improves the performance of the segmentation network.

   As shown in Fig. 3, while the proposed DASGAN model systematically outperforms the baseline model trained only on real PD-L1 samples, the relative improvement in accuracy metrics tends to decrease with the availability of manual annotations. In the configuration (iii) of highest availability, accuracy metrics of $f_1 = (0.894/0.890)$ and $f_1 = (0.916/0.899)$ are reached by the baseline model trained only on real PD-L1 samples and by our approach respectively. This quantitatively confirms the expectation that the use of synthetic data is most relevant in case of relative shortage of manually labeled data.

## 3.4   Tumor Cell scoring

PD-L1 status, which is predictive for survival of NSCLC patients receiving PD1/PD-L1 checkpoint inhibitor therapy [11], is determined based on the Tu-

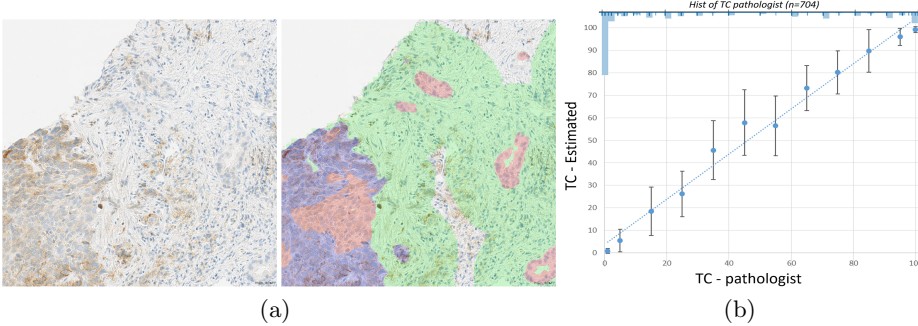

(a)                                                                                    (b)

**Fig. 4.** (a) Example of negative (red) and positive (blue) epithelial regions as well as of non-epithelial regions (green) segmented by the proposed DASGAN model. (b) Bar plot showing mean and standard deviation of the TC scores estimated by the proposed approach on unseen cases (n=704), on the following TC score bins: $TC < 1$, $1 <= TC < 10$, $10n <= TC < 10(n+1)$ for $1 < n < 10$ and $TC = 100$. The inverted histogram at the top shows the distribution of cases w.r.t. pathologist TC score.

mor Cell score, defined as the percentage of tumor epithelial cells that are PD-L1 positive. Following [6], the TC score is approximated as the relative area of the detected TC(+) regions:

$$TC_{CNN} = \frac{\#TC(+)}{\#TC(-) + \#TC(+)}. \tag{6}$$

Fig. 4a displays an example of epithelial segmentation output by the proposed DASGAN model. To quantitatively assess the clinical relevance of the proposed approach, we consider a set of 704 PD-L1 stained images unseen for training nor selection of the segmentation model. This set originates from three independent patient cohorts and contain both needle biopsies and resectates. Fig. 4b shows the bar plot of mean and standard deviation of the estimated $TC_{CNN}$ scores against the true TC scores visually estimated by pathologists. Lin's concordance coefficient of $Lcc = 0.93$, Pearson correlation coefficient of $Pcc = 0.94$ and mean absolute error of $MAE = 7.30$ are reported between the estimated and the true TC score values, quantitatively showing the high concordance of the proposed method with visual scoring by pathologist.

## 4 Discussion and Conclusion

In this paper, we introduce a novel method to leverage data from two stain domains (CK and PD-L1) and two independent cohorts for the segmentation of epithelium in PD-L1 images. The semi-automatic generation of large boundary-precise datasets for epithelium segmentation in CK images together with their unpaired translation into realistic-looking images PD-L1 images makes it possible to generate large dataset for epithelial segmentation in the PD-L1 stain domain, without the need for serial sections or re-staining of slides.

The proposed DASGAN model performs joint domain translation and semantic segmentation. As experimentally shown, it enables (i) the segmentation of the epithelial regions, (ii) the segmentation of PD-L1 positive and negative epithelial regions, (iii) the replication of the clinically relevant PD-L1 Tumor Cell (TC) score. Upon confirmation that the presented results match survival predictive ability of manual scoring and replication of the findings in a prospective trial, we envision that the proposed method could be used in a clinical setting to identify patients which may benefit from an anti-PD1/PD-L1 therapy. A direct extension of this work would the analysis of slides stained with HE and other immunohistochemistry markers. More generally, we believe that the proposed joint domain adaptation and segmentation methodology is very generic and can be applied to the analysis of a wide variety of histopathological data.

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
