# OpenReview forum: "DASGAN - Joint Domain Adaptation and Segmentation for the Analysis of Epithelial Regions in Histopathology PD-L1 Images"
_MICCAI.org/2019/Workshop/COMPAY — COMPAY 2019_

### Official Review · AnonReviewer1 · 2019-07-30
**Interesting domain adaption, and learning of segmentation with different staining.**

**Rating:** 8
**Confidence:** 4

**Review:**

The authors present an interesting GAN constellation to learn Epithelium segmentation in PD-L1 stained lung images with the help of CK stained images. This method allows for learning from two different staining which can be un-related (no consecutive slices, no re-staining, etc..).

Although the principal concept of domain transfer is not new (the authors of course also mention CT to MRI, or HE to normalization transfer), their method jointly performs domain translation AND segmentation. I think this is an interesting application in computational pathology.

The paper is well written and structured. However, I recommend a few minor improvements:

- The validation approach should be clarified better. Very good is the validation in two directions: The segmentation masks itself, and the downstream Tumor Cell Score achieved with the segmentations. However, the segmentation validation comes in different flavors: with few annotations (config (i)) and many annotations (conf (ii)). What is the exact difference between Fig 2 and Fig 3? Should Fig 2 be in the range of Fig 3 config (i) (few annotations)?

- In your test set, when calculating the Tumor Cell Score of PD-L1 without any mask at all, is that worse than using Epithelial Cell masks? In other words, is your method (and Epithelial segmentation) really needed in the test set, or is the test set so selected such that is comprises Epithelial cells only anyway? A good test set would consist of many different areas (necrotic, epithelial, immune, ...) such that the Tumor Cell Score would be really screwed up without a proper Epithelial segmentation. To show this, I would recommend to compare the TC Score with a PD-L1 score without any mask. In the best case, the correlation should be low.

- Please state the accuracy in the summary.

---

### Official Review · AnonReviewer4 · 2019-08-15

**Rating:** 7
**Confidence:** 4

**Review:**

This paper makes an extension of the existing cyclegan framework for segmentation purpose. I think it is an interesting idea to make use of cyclegan for segmentation purpose.
A few works on cyclegan -based segmentation is already conducted such as https://arxiv.org/abs/1905.01902 , please consider to discuss.
It is not clear how does this method compare to the scheme that train CK with one domain and fine-tune on the other domain
Please also evaluate the quality of the domain adaption.

---

### Decision · Program_Chairs · 2019-08-20

Accept